# On Non-Random Missing Labels in
# Semi-Supervised Learning

**Xinting Hu**[1]   **Yulei Niu**[1]*   **Chunyan Miao**[1]   **Xian-Sheng Hua**[2]   **Hanwang Zhang**[1]
[1]Nanyang Technological University, [2]Damo Academy, Alibaba Group
`xinting001@e.ntu.edu.sg, yn.yuleiniu@gmail.com`
`{ascymiao, hanwangzhang}@ntu.edu.sg, xiansheng.hxs@alibaba-inc.com`

## ABSTRACT

Semi-Supervised Learning (SSL) is fundamentally a missing label problem, in which the label Missing Not At Random (MNAR) problem is more realistic and challenging, compared to the widely-adopted yet naïve Missing Completely At Random assumption where both labeled and unlabeled data share the same class distribution. Different from existing SSL solutions that overlook the role of "class" in causing the non-randomness, *e.g.*, users are more likely to label popular classes, we explicitly incorporate "class" into SSL. Our method is three-fold: 1) We propose Class-Aware Propensity (CAP) score that exploits the unlabeled data to train an improved classifier using the biased labeled data. 2) To encourage rare class training, whose model is low-recall but high-precision that discards too many pseudo-labeled data, we propose Class-Aware Imputation (CAI) that dynamically decreases (or increases) the pseudo-label assignment threshold for rare (or frequent) classes. 3) Overall, we integrate CAP and CAI into a Class-Aware Doubly Robust (CADR) estimator for training an unbiased SSL model. Under various MNAR settings and ablations, our method not only significantly outperforms existing baselines, but also surpasses other label bias removal SSL methods.

## 1 INTRODUCTION

Semi-supervised learning (SSL) aims to alleviate the strong demand for large-scale labeled data by leveraging unlabeled data (Zhu, 2008; Yang et al., 2021). Prevailing SSL methods first train a model using the labeled data, then uses the model to *impute* the missing labels with the predicted pseudo-labels for the unlabeled data (Van Buuren, 2018), and finally combine the true- and pseudo-labels to further improve the model (Sohn et al., 2020; Berthelot et al., 2019a). Ideally, the "improve" is theoretically guaranteed if the missing label imputation is perfect (Grandvalet & Bengio, 2005); otherwise, imperfect imputation causes the well-known *confirmation bias* (Arazo et al., 2019; Sohn et al., 2020). In particular, the bias is even more severe in practice because the underlying assumption that the labeled and unlabeled data are drawn from the same distribution does not hold. We term this scenario as label Missing Not At Random (MNAR) (Hernán & Robins, 2020), as compared to the naïve assumption called label Missing Completely At Random (MCAR).

MNAR is inevitable in real-world SSL due to the limited label annotation budget (Rosset et al., 2005)—uniform label annotations that keep MCAR are expensive. For example, we usually deploy low-cost data collecting methods like crawling social media images from the Web (Mahajan et al., 2018). However, the high-quality labels appear to be severely imbalanced over classes due to the imbalanced human preferences for the "class" (Misra et al., 2016; Colléony et al., 2017). For example, people are more willing to tag a "Chihuahua" rather than a "leatherback" (a huge black turtle) since the former is more lovely and easier to recognize. Thus, the imputation model trained by such biased supervision makes the model favoring "Chihuahua" rather than "leatherback". The "Chihuahua"-favored pseudo-labels, in turn, negatively reinforce the model confidence in the false "Chihuahua" belief and thus exacerbate the confirmation bias.

We design an experiment on CIFAR-10 to further illustrate how existing state-of-the-art SSL methods, *e.g.*, FixMatch (Sohn et al., 2020), fail in the MNAR scenario. As shown in Figure 1(a), the

---

*Now in Columbia University

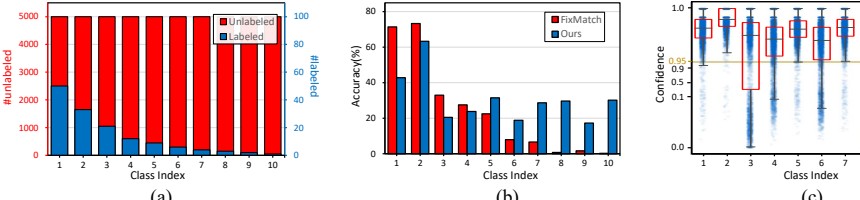

Figure 1: Visualization of an MNAR example and its experimental results on CIFAR-10. (a) Class distribution of the labeled and unlabeled training data. (b) Test accuracy of the supervised model using FixMatch and our CAP (Section 4.1). (c) The distribution of FixMatch's confidence scores on unlabeled data. Samples with confidence larger than a fixed threshold (0.95, yellow line) are imputed. The corresponding box-plots display the (minimum, first quartile, median, third quartile, maximum) summary. The confidence-axis is not equidistant scaling for better visualization.

overall training data is uniformly distributed over classes, but the labeled data is long-tailed distributed, which simulates the imbalanced class popularity. Trained on such MNAR training data, FixMatch even magnifies the bias towards the popular classes and ignores the rare classes (Figure 1(b)). We point out that the devil is in the imputation. As shown in Figure 1(c), the confidence scores of FixMatch for popular classes are much higher than those for rare classes (*i.e.*, averagely >0.95 vs. <0.05). Since FixMatch adopts a fixed threshold for the imputation samples selection, *i.e.*, only the samples with confidence larger than 0.95 are imputed, FixMatch tends to impute the labels of more samples from the popular classes , which still keeps the updated labels with the newly added pseudo-labels long-tailed. As a result, FixMatch is easily trapped in MNAR.

In fact, MNAR has attracted broad attention in statistics and Inverse Propensity Weighting (IPW) is one of the commonly used tools to tackle this challenge (Seaman & White, 2013; Jones et al., 2006). IPW introduces a weight for each training sample via its propensity score, which reflects how likely the label is observed (*e.g.*, its popularity). In this way, IPW makes up a pseudo-balanced dataset by duplicating each labeled data inversely proportional to its propensity—less popular samples should draw the same attention as the popular ones—a more balanced imputation. To combine the IPW true labels and imputed missing labels, a Doubly Robust (DR) estimator is used to guarantee a robust SSL model if either IPW or imputation is unbiased (Seaman & Vansteelandt, 2018; Vansteelandt et al., 2010).

However, the above framework overlooks the role of "class" in causing the MNAR problem of SSL (Section 3), and we therefore propose a unified Class-Aware Doubly Robust (CADR) framework to address this challenge (Section 4). CADR removes the bias from both the supervised model training and the unlabeled data imputation ends. For the former end, we propose Class-Aware Propensity (CAP) that exploits the unlabeled data to train an improved classifier using the biased labeled data (Section 4.1). We use an efficient moving averaging strategy to estimate the propensity. For the latter end, we design Class-Aware Imputation (CAI) that dynamically adjusts the pseudo-label assignment threshold with class-specific confidence scores to mitigate the bias in imputation (Section 4.2). Intuitively, CAI lowers the threshold requirement of rare class samples to balance with frequent class samples. Experiments on several image classification benchmarks demonstrate that our CADR framework achieves a consistent performance boost and can effectively tackle the MNAR problem. Besides, as MCAR is a special case of MNAR, our method maintains the competitive performance in the conventional SSL setting.

The key contributions of this work are summarized as follows:

- We propose a realistic and challenging label Missing Not At Random (MNAR) problem for semi-supervised learning, which is not extensively studied in previous work. We systematically analyze the bias caused by non-random missing labels.

- We proposes a unified doubly robust framework called Class-Aware Doubly Robust (CADR) estimator to remove the bias from both the supervised model training end—by using Class-Aware Prospesity (CAP), and the unlabeled data imputation end—by using Class-Aware Imputation (CAI).

- Our proposed CADR achieves competitive performances in both MNAR and conventional label Missing Completely At Random (MCAR).

## 2   RELATED WORKS

**Missing Not At Random**. Missing data problems are ubiquitous across the analysis in social, behavioral, and medical sciences (Enders, 2010; Heckman, 1977). When data is not missing at random, estimations based on the observed data only results in bias (Heckman, 1977). The solutions are three-fold: 1) *Inverse probability weighting* (IPW) assigns weights to each observed datum based on the propensity score, *i.e.*, the probability of being observed. A missing mechanism-based model is used to estimate the sample propensity, such as a logistic regression model (Rosset et al., 2005; Wang et al., 1997) or a robit regression model (Kang & Schafer, 2007; Liu, 2005). 2) *Imputation* methods aim to fill in the missing values to produce a complete dataset (Kenward & Carpenter, 2007; Little & Rubin, 2014). An imputation model is regressed to predict the incomplete data from the observed data in a deterministic (Kovar & Whitridge., 1995) or stochastic (Lo et al., 2019) way. 3) As the propensity estimation or the regressed imputation are easy to be biased, *doubly robust* estimators propose to integrate IPW and imputation with double robustness (Seaman & Vansteelandt, 2018; Vansteelandt et al., 2010; Jonsson Funk et al., 2011): the capability to maintain unbiased when either the propensity or imputation model is biased. In this work, as the image labels are missing not at random, we design the class-aware propensity and imputation considering the causal role of label classes in missing, avoiding the uncertain model specification and generating unbiased estimation. These two class-aware modules can be naturally integrated into a doubly robust estimator.

**Semi-Supervised Learning (SSL)**. It aims to exploit unlabeled data to improve the model learned on labeled data. Prevailing SSL methods (Sohn et al., 2020; Berthelot et al., 2019b;a) share a similar strategy: training a model with the labeled data and generating pseudo-labels for unlabeled data based on the model predictions. Pseudo-labeling methods (Lee, 2013; Xie et al., 2020; Rizve et al., 2021) predict pseudo-labels for unlabeled data and add them to the training data for re-training. Consistency-regularization methods (Sajjadi et al., 2016; Laine & Aila, 2016; Berthelot et al., 2019b) apply a random perturbation to an unlabeled image and then use the prediction as the pseudo-label of the same image under a different augmentation. Recent state-of-the-art SSL methods (Sohn et al., 2020; Berthelot et al., 2019a) combine the two existing techniques and predict improved pseudo-labels. However, when labels are missing at random, these methods can be inefficient as the model learned on labeled data is biased and significantly harms the quality of pseudo labels. Though some works also notice the model bias (Wei et al., 2021; Kim et al., 2020), they neglect the causal relationship between the bias and the missing label mechanism in SSL, which is systematically discussed in our work.

## 3   MISSING LABELS IN SEMI-SUPERVISED LEARNING

In semi-supervised learning (SSL), the training dataset $D$ is divided into two disjoint sets: a labeled dataset $D_L$ and an unlabeled data $D_U$. We denote $D_L$ as $\{(x^{(i)}, y^{(i)})\}_{i=1}^{N_L}$ (usually $N_L \ll N$) where $x^{(i)} \in \mathbb{R}^d$ is a sample feature and $y^{(i)} \in \{0, 1\}^C$ is its one-hot label over $C$ classes, and $D_U$ as $\{(x^{(i)})\}_{i=N_L+1}^{N}$ where the remaining $N - N_L$ labels are missing in $D_U$. We further review SSL as a label missing problem, and define a *label missing indicator* set $M$ with $m^{(i)} \in \{0, 1\}$, where $m^{(i)} = 1$ denotes the label is missing (*i.e.*, unlabeled) and $m^{(i)} = 0$ denotes the label is not missing (*i.e.*, labeled). In this way, we can rewrite the dataset as $D = (X, Y, M) = \{(x^{(i)}, y^{(i)}, m^{(i)})\}_{i=1}^{N}$.

Traditional SSL methods assume that $M$ is independent with $Y$, *i.e.*, the labels are Missing Completely At Random (MCAR). Under this assumption, prevailing SSL algorithms can be summarized as the following multi-task optimization of the supervised learning and the unlabeled imputation:

$$\hat{\theta} = \arg\min_{\theta} \sum_{(x,y) \in D_L} \mathcal{L}_s(x, y; \theta) + \sum_{x \in D_U} \mathcal{L}_u(x; \theta), \tag{1}$$

where $\mathcal{L}_s$ and $\mathcal{L}_u$ are the loss function designed separately for supervised learning on labeled data ($M = 0$) and regression imputation on unlabeled data ($M = 1$). In general, SSL methods first train a model with parameters $\hat{\theta}$ using $\mathcal{L}_s$ on $D_L$, then *impute* the missing labels (Van Buuren, 2018) by predicting the pseudo-labels $\hat{y} = f(x; \hat{\theta})$, and finally combine the true- and pseudo-labels to further improve the model for another SSL loop, *e.g.*, another optimization iteration of Eq. (1). The implementation of $\mathcal{L}_s$ and $\mathcal{L}_u$ are open. For example, $\mathcal{L}_s$ is normally the standard cross-entropy

loss, and $\mathcal{L}_u$ can be implemented as squared $L_2$ loss (Berthelot et al., 2019b) or cross-entropy loss (Berthelot et al., 2019a; Sohn et al., 2020).

As you may expect, the key to stable SSL methods is the unbiasedness of the imputation model. When $M$ is independent with $Y$, we have $P(Y|X=x, M=0) = P(Y|X=x)$, and thus

$$
\begin{aligned}
\mathbb{E}[\hat{y}] = \mathbb{E}[y|\hat{\theta}] &= \sum_{(x,y)\in D_L} y \cdot P(y|x) = \sum_{(x,y)\in D} y \cdot P(y|x, M=0) \\
&= \sum_{(x,y)\in D} y \cdot P(y|x) = \mathbb{E}[y],
\end{aligned}
\tag{2}
$$

which indicates that the model is an ideally unbiased estimator, and the imputed labels are unbiased. However, such MCAR assumption is too strong and impractical. For example, annotators may tend to tag a specific group of samples due to their preferences or experiences. This motivates us to ask the following realistic but overlooked question: *what if $M$ is dependent with $Y$, i.e.*, the labels are missing not at random (MNAR)? In this case, we have $P(Y|X=x, M=0) \neq P(Y|X=x)$, and thus

$$
\begin{aligned}
\mathbb{E}[\hat{y}] &= \sum_{(x,y)\in D} y \cdot P(y|x, M=0) = \sum_{(x,y)\in D} y \cdot P(y|x) \cdot \frac{P(M=0|x,y)}{P(M=0|x)} \\
&\neq \sum_{(x,y)\in D} y \cdot P(y|x) = \mathbb{E}[y].
\end{aligned}
\tag{3}
$$

We can see it is no longer unbiased, further leading to biased imputed labels. As a result, the bias will be increasingly enhanced with the joint optimization loop in Eq. (1) moving forward. This analysis indicates that previous SSL methods are not applicable for the MNAR problem as they overlook the role of "class" $Y$ in causing the non-randomness of $M$. In the following, we will introduce our Class-Aware Doubly Robust framework that targets at an unbiased estimator.

## 4   CLASS-AWARE DOUBLY ROBUST FRAMEWORK

As discussed above, both the supervised learning and regression imputation in previous SSL methods suffer from the biased label estimation for MNAR. In this section, we proposed a doubly robust estimator that explicitly incorporates "class" into SSL and mitigates the bias from both sides.

### 4.1   CLASS-AWARE PROPENSITY FOR LABELED DATA

Traditional SSL methods estimate the model parameters via maximum likelihood estimation on the labeled data:

$$
\hat{\theta} = \arg\max_{\theta} \log P(Y|X, M=0; \theta) = \arg\max_{\theta} \sum_{(x,y)\in D_L} \log P(y|x; \theta)
\tag{4}
$$

Since $\hat{\theta}$ is estimated over $D_L$ rather than the entire $D$, the difference between $D_L$ and $D$ determines the unbiasedness of $\hat{\theta}$. As analyzed in Section 3, $\hat{\theta}$ is an unbiased estimator if the labels $Y$ and their missing indicators $M$ are independent, *i.e.*, MCAR. However, when $M$ is dependent with $Y$, *i.e.*, MNAR, we have $P(y|x; M=0) \neq P(y|x)$. In this case, $\hat{\theta}$ is no longer unbiased.

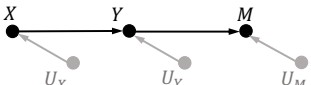

Figure 2: Structural Causal Model for MNAR. $U_X$, $U_Y$ and $U_M$ denote the exogenous variables.

Inspired by causal inference (Pearl et al., 2016), we establish a structural causal model for MNAR to analyze the causal relations between $X$, $Y$ and $M$ shown in Figure 2. First, $X \to Y$ denotes that each image determines its own label. Second, $Y \to M$ denotes that the missing of label is dependent with its category. For example, popular pets like cat and dog attracts people to tag more often. We observe a chain structure of $X \to Y \to M$, which obtains an important conditional independence rule (Pearl et al., 2016): Variables $A$ and $B$ are conditionally independent given $C$, if there is only one unidirectional path between $A$ and $B$, and $C$ is any set of variables intercepting the path. Based on this rule, we have

$P(X|Y, M = 0) = P(X|Y)$ when conditioning on $Y$, *i.e.*, the path between $X$ and $M$ is blocked by $Y$. In this way, we can obtain the unbiased $\hat{\theta}$ on the labeled subset of data by maximizing $P(X|Y)$:

$$\hat{\theta} = \arg\max_{\theta} \log P(X|Y; \theta) = \arg\max_{\theta} \sum_{(x,y) \in D_L} \log P(x|y; \theta) \tag{5}$$

$$= \arg\max_{\theta} \sum_{(x,y) \in D_L} \log \frac{P(y|x; \theta)P(x; \theta)}{P(y; \theta)} \tag{6}$$

$$= \arg\max_{\theta} \sum_{(x,y) \in D_L} \log \frac{P(y|x; \theta)}{P(y; \theta)} \tag{7}$$

$$= \arg\max_{\theta} \sum_{(x,y) \in D_L} \log P(y|x; \theta) \cdot \frac{\log P(y|x; \theta) - \log P(y; \theta)}{\log P(y|x; \theta)} \tag{8}$$

$$\triangleq \arg\max_{\theta} \sum_{(x,y) \in D_L} \log P(y|x; \theta) \cdot \frac{1}{s(x, y)}. \tag{9}$$

Eq. (5) to Eq. (6) holds by Bayes' rule; Eq. (6) to Eq. (7) holds because $x$ is sampled from an empirical distribution and $p(x; \theta)$ is thus constant. Eq.(7) to Eq.(9) shows the connection with Inverse Probability Weighting, where we estimate the propensity score $s(x, y)$ for image $x$ as $\frac{\log P(y|x; \theta)}{\log P(y|x; \theta) - \log P(y; \theta)}$. Comparing the training objectives in Eq. (9) with Eq. (4), our estimation can be understood as adjusting the original $P(y|x; \theta)$ by a class-aware prior $P(y; \theta)$ for each labeled data $(x, y)$.

To estimate $P(y; \theta)$, *i.e.*, the marginal distribution $P(Y = y)$ with the current parameters $\theta$, a straightforward solution is to go through the whole dataset and calculate the mean probability of all the data. However, during the training stage, $\theta$ is updated step by step, and it is impractical to go thorough the whole dataset in each iteration considering the memory and computational cost. An alternative solution is to estimate the propensity within a mini-batch. Although the cost is saved, it faces a risk of high-variance estimation.

To resolve the dilemma, we propose to use a moving averaging strategy over all the mini-batches. Specifically, we keep a buffer $\hat{P}(Y)$ to estimate $P(Y; \theta_t)$ and continuously update it by $P(Y; B_t, \theta_t)$, which is estimated with the current mini-batch of $B_t$ samples and parameters $\theta_t$ at $t$-th iteration:

$$\hat{P}(Y) \leftarrow \mu\hat{P}(Y) + (1 - \mu)P(Y; B_t, \theta_t), \tag{10}$$

where $\mu \in [0, 1)$ is a momentum coefficient.

## 4.2 CLASS-AWARE IMPUTATION FOR UNLABELED DATA

As discussed in Section 1, the imputation model in traditional SSL is biased towards popular classes, and so are the imputed labels. Although our proposed CAP can mitigate the preference bias during the supervised learning stage, the model can still tend to impute popular class that achieves a high confidence score. To further alleviate the bias in the imputation stage, we propose to remove the inferior imputed labels from $\hat{Y}$. Traditional models like FixMatch adopt a fixed threshold for imputation samples selection, which aims to reserve the accurate imputed labels and discard the noisy ones. However, as shown in Figure 1 (c), for specific classes, an extremely small number of imputed labels can meet the requirement while most are discarded. In fact, as observed in previous works (Wei et al., 2021), these discarded predictions still hold nearly perfect precision. Based on these observations, we believe that a fixed threshold is too coarse to impute missing labels.

To tackle this challenge, we propose a Class-Aware Imputation (CAI) strategy that dynamically adjusts the pseudo-label assignment threshold for different classes. Let $C_x$ denote the potential imputed label for image $x$, *i.e.*, $C_x = \arg\max_y P(y|x; \theta)$. We use a class-aware threshold $\tau(x)$ for image $x$ as:

$$\tau(x) = \tau_o \cdot \left(\frac{\hat{P}(C_x)}{\max_{y \in \{1, \cdots, C\}} \hat{P}(y)}\right)^{\beta}, \tag{11}$$

where $\tau_o$ is the conventional threshold, $\beta$ is a hyper-parameter, and $\hat{P}(y)$ is the class-aware propensity for class $y$ estimated by CAP. Intuitively, our class-aware threshold would set a higher requirement for popular classes and a lower requirement for rare classes, allowing more samples from rare

classes to be imputed. With the gradual removal of bias from the training process, the performance gap between classes also shrinks, and both popular and rare classes can be fairly treated. Furthermore, when labels are missing completely at random (MCAR), our CIA can be degraded to the conventional imputation like previous works, as $\hat{P}(Y)$ is uniform over all classes.

### 4.3 CLASS-AWARE DOUBLY ROBUST ESTIMATOR

Note that CAP and CAI mitigate the bias in supervised learning and imputation regression. Thanks to the theory of double robustness (Seaman & Vansteelandt, 2018; Vansteelandt et al., 2010), we can incorporate CAP and CAI into a Class-Aware Doubly Robust (CADR) estimator. Theoretically, the CADR combination has a lower tail bound than applying each of the components alone (Wang et al., 2019). Besides, it provides our learning system with double robustness: the capability to remain unbiased if either CAP or CAI is unbiased.

Following the formulation of DR estimator, we first rewrite the training objective of CAP and CAI in semi-supervised learning as:

$$\mathcal{L}_{\text{CAP}} = \frac{1}{N} \sum_{i=1,\cdots,N} \frac{(1 - m^{(i)})\mathcal{L}_s(x^{(i)}, y^{(i)})}{p^{(i)}} \tag{12}$$

$$\mathcal{L}_{\text{CAI}} = \frac{1}{N} \sum_{i=1,\cdots,N} (m^{(i)}\mathcal{L}_u(x^{(i)}, q^{(i)})\,\mathbb{I}(\text{con}(q^{(i)}) > \tau(x^{(i)})) + (1 - m^{(i)})\mathcal{L}_s(x^{(i)}, y^{(i)})), \tag{13}$$

where $m^{(i)}$ is the missing state, $p^{(i)}$ is the propensity score, $q^{(i)}$ is the imputed label with confidence $\text{con}(q^{(i)})$, and $\mathbb{I}(\cdot)$ is the indicator function. As introduced in Section 4.1, we estimate the propensity score $p^{(i)}$ as $s(x^{(i)}, y^{(i)})$ in CAP. Then the optimization of CADR estimator is implemented as

$$\hat{\theta}_{\text{CADR}} = \arg\min_{\theta} \mathcal{L}_{\text{CADR}} = \arg\min_{\theta} \mathcal{L}_{\text{CAP}} + \mathcal{L}_{\text{CAI}} + \mathcal{L}_{\text{supp}}, \tag{14}$$

$$\text{where} \qquad \mathcal{L}_{\text{supp}} = \frac{1}{N} \sum_{i=1,\cdots,N} (1 - m^{(i)} - \frac{1 - m^{(i)}}{p^{(i)}})\mathcal{L}_u(x^{(i)}, q^{(i)})\,\mathbb{I}(\text{con}(q^{(i)}) > \tau) \tag{15}$$

$$- \frac{1}{N} \sum_{i=1,\cdots,N} (1 - m^{(i)})\mathcal{L}_s(x^{(i)}, y^{(i)}),$$

which is a supplementary loss to guarantee the unbiasedness. In this design, $\mathcal{L}_{\text{CAI}} + \mathcal{L}_{\text{supp}}$ is expected to be 0 given correct CAP, and $\mathcal{L}_{\text{CAP}} + \mathcal{L}_{\text{supp}}$ is expected to be 0 given correct CAI. These results guarantee the double robustness in case that either the propensity or imputation is inaccurate.

## 5 EXPERIMENTS

### 5.1 EXPERIMENTAL SETUP

**Datasets.** We evaluate our method on four image classification benchmark datasets: CIFAR-10, CIFAR-100 (Krizhevsky, 2012), STL-10 (Coates et al., 2011) and mini-ImageNet (Vinyals et al., 2016). CIFAR-10(-100) is composed of 60,000 images of size $32 \times 32$ from 10 (100) classes and each class has 5,000 (500) training images and 1,000 (100) samples for evaluation. STL-10 dataset has 5,000 labeled images and 100,000 unlabeled images of size $64 \times 64$. mini-ImageNet is a subset of ImageNet (Deng et al., 2009). It contains 100 classes where each class has 600 images of size $84 \times 84$. Follows previous SSL works (Hu et al., 2021; Iscen et al., 2019), we select 500 images from each class for training and 100 images per class for testing.

**MNAR Settings.** Since the training data is class-balanced in all datasets, we randomly select a class-imbalanced subset as the labeled data to mimic the label missing not at random (MNAR). For a dataset containing $C$ classes, the number of labeled data in each class $N_i$ are calculated as $N_i = N_{max} \cdot \gamma^{-\frac{i-1}{C-1}}$. $N_1 = N_{max}$ is the maximum number of labeled data among all the classes, and $\gamma$ describes the imbalance ratio. $\gamma = 1$ when the labeled data is balanced over classes, and larger $\gamma$ indicates more imbalanced class distribution. Figure 1(a) shows an example of the data distribution when $\gamma = N_{max} = 50$ in CIFAR-10. We also consider other MNAR settings where the unlabeled data is also imbalanced, *i.e.*, the imbalance ratio of unlabeled data $\gamma_u$ does not equal to 1.

To ensure the number of unlabeled data is much larger than the labeled, we set the least number of labeled data over all classes as 1, and the largest number of unlabeled data to its original size.

**Training Details.** Following previous works (Berthelot et al., 2019b; Sohn et al., 2020; Hu et al., 2021), we used Wide ResNet (WRN)-28-2 for CIFAR-10, WRN-28-8 for CIFAR-100, WRN-37-2 for STL-10 and ResNet-18 for mini-Imagenet. Since our methods are implemented as a plug-in module to FixMatch, common network hyper-parameters, *e.g.*, learning rates, batch-sizes, are the same as their original settings (Sohn et al., 2020). For each dataset, our model and FixMatch are trained $2^{17}$ iterations in MNAR and $2^{20}$ steps in ordinary SSL cases ($\gamma = 1$).

| Method | CIFAR-10 | | | CIFAR-100 | | | STL-10 | | mini-ImageNet | |
|---|---|---|---|---|---|---|---|---|---|---|
| | $\gamma = 20$ | 50 | 100 | 50 | 100 | 200 | 50 | 100 | 50 | 100 |
| Π Model | 21.59 | 27.54 | 30.39 | 24.95 | 29.93 | 33.91 | 31.89 | 34.69 | 11.77 | 15.30 |
| MixMatch | 26.63 | 31.28 | 28.02 | 37.82 | 41.32 | 42.92 | 28.98 | 28.31 | 13.12 | 18.30 |
| ReMixMatch | 41.84 | 38.44 | 38.20 | 42.45 | 39.71 | 39.22 | 41.33 | 39.55 | 22.64 | 23.50 |
| FixMatch | 56.26 | 65.61 | 72.28 | 50.51 | 48.82 | 50.62 | 47.22 | 57.01 | 23.56 | 26.57 |
| + CREST | 51.10 | 55.40 | 63.60 | 40.30 | 46.30 | 49.60 | – | – | – | – |
| + DARP | 63.14 | 70.44 | 74.74 | 38.87 | 40.49 | 44.15 | 39.66 | 39.72 | – | – |
| + CADR (Ours) | **79.63** | **93.79** | **93.97** | **59.53** | **60.88** | **63.30** | **70.29** | **76.70** | **29.07** | **32.78** |

Table 1: A Comparison of mean accuracies (%). We alter the imbalance ratio $\gamma$ of labeled data and leave the unlabeled data balanced ($\gamma_u = 1$). We keep $N_{max} = \gamma$ so that the least number of labeled data among all the classes is always 1.

## 5.2 RESULTS AND ANALYSES

**Comparison with State-of-The-Art Methods.** To demonstrate the effectiveness of our method, we compared it with multiple baselines using the same network architecture, including CREST (Wei et al., 2021) and DARP (Kim et al., 2020) that aims to handle unbalanced semi-supervised learning. All the methods are implemented based on their public codes. Table 1 shows the results on all the datasets with different levels of imbalanced label ratios. While CREST and DARP fail to correct the severely biased learning process, our method outperforms all the baselines by large margins across all the settings. The reason is that though CREST and DARP handle the unbalanced labeled data, they mainly focus on the case where the unlabeled data is equally unbalanced. As the labeled and the unlabeled data share the same class distribution, the pseudo-labels they used are not as misleading as in our cases. Thus the bias removal is not critically emphasized by their algorithms. On CIFAR-10 and CIFAR-100, we boost the mean accuracy of FixMatch by 24.41% and 11.26% on average. On STL-10 and more challenging mini-ImageNet, our improvements are also substantial (21.38% and 5.86%). Specifically, Figure 3 shows the convergence trend and confusion matrices under one MNAR scenario ($\gamma = N_{max} = 50$). As one can observe, the performance of FixMatch is boosted mainly due to the de-biased estimation for the labeled-data-rare classes.

**Individual Effectiveness of Each Component.** Table 2 shows the results of using Class-Aware Propensity (CAP) and Class-Aware Imputation (CAI) alone and together in trivial ($\mathcal{L}_{CAP} + \mathcal{L}_{CAI}$, w/o CADR) and our DR combination ($\mathcal{L}_{CAP} + \mathcal{L}_{CAI} + \mathcal{L}_{supp}$, w/ CADR). We can observe that the improvement of using either CAP and CAI alone is distinguishable, demonstrating that both CAP and CAI achieve bias-removed estimation. Combining CAI and CAP, our CADR exhibits

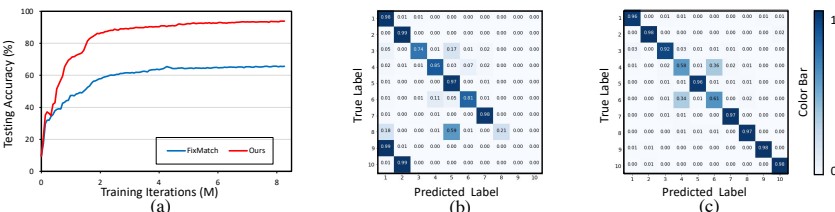

Figure 3: (a): the convergence trends of FixMatch and our method. (b) and (c): the confusion matrices of FixMatch and ours. Results are on CIFAR-10 ($\gamma = N_{max} = 50$).

| Method | CIFAR-10 | | | CIFAR-100 | | | STL-10 | | mini-ImageNet | |
|---|---|---|---|---|---|---|---|---|---|---|
| | $\gamma$= 20 | 50 | 100 | 50 | 100 | 200 | 50 | 100 | 50 | 100 |
| FixMatch | 56.26 | 65.61 | 72.28 | 50.51 | 48.82 | 50.62 | 47.22 | 57.01 | 23.56 | 26.57 |
| w/ CAP | 79.38 | 89.50 | 93.95 | 55.72 | 58.53 | 63.07 | 69.97 | **77.69** | 28.54 | 32.23 |
| w/ CAI | 79.02 | 88.15 | 93.86 | 58.55 | 59.80 | 58.26 | **70.64** | 71.21 | 27.15 | 30.94 |
| w/o CADR | 79.43 | 89.53 | **94.10** | 57.93 | 59.50 | 62.78 | 70.17 | 75.44 | 25.54 | 31.66 |
| w/ CADR | **79.63** | **93.79** | 93.97 | **59.53** | **60.88** | **63.30** | 70.29 | 76.70 | **29.07** | **32.78** |

Table 2: The individual performance of our proposed Class-Aware Propensity (CAP) and Class-Aware Imputation (CAI) alone and together in trivial combination (w/o CADR) and CADR combination (w/ CADR). We marked the **best** and second-best accuracies.

steady improvement over the baseline. We found that our CADR is consistently among the best or second-best performance, agnostic to the value of $\gamma$ and the performance gap between individual components. When the label data is more imbalanced, *e.g.*, $\gamma = 100$, CAP outperforms CAI by large margins on CIFAR-100 (63.07% *vs.* 58.26%), STL-10 (77.69% *vs.* 71.21%), and mini-ImageNet (32.23% *vs.* 30.94%), and our CADR outperforms the trivial combination by 0.52% $\sim$ 1.26%. The special case is $\gamma = 100$ on CIFAR-10, where the trivial combination slightly beats CADR (93.97% *vs.* 94.10%), and the gap between CAP and CAI is also small (93.96% *vs.* 93.86%). These results empirically verified our theoretical analysis and the robustness of our CADR, that is , CADR maintains unbiasedness when either CAP or CAI is unbiased. To sum up, CADR can provide a reliable solution with robust performance and bring convenience to real applications.

**Comparison with More Baselines.** Apart from CREST and DARP, we further compare with some re-balancing techniques adopted in long-tailed fully-supervised learning in Table 3. They are 1) *re-weighting* (Cui et al., 2019): re-weighting labeled sample according to the inverse of the number of labeled data in each class, 2) *re-sampling*: re-sampling the labeled data to construct a balanced labeled set, and 3) *LA* (Menon et al., 2020): using logits adjusted softmax cross-entropy loss that applies a label-dependent offset to logits for each labeled sample. However, under MNAR, where the model is biased by supervised learning and unlabeled data imputation together, only dealing with the supervised process without considering the whole data distribution gives inferior results. A recent work, DASH (Xu et al., 2021) also proposed to use a dynamic threshold in filtering imputed labels. Compared to their work, our threshold is both dynamic and class-aware, and our CADR outperforms DASH by a large margin.

| Method | Accuracy (%) |
|---|---|
| FixMatch | 65.61 |
| + CREST | 55.40 (-10.21) |
| + DARP | 70.44 (+4.83) |
| + re-weighting | 66.03 (+0.42) |
| + re-sampling | 65.49 (-0.12) |
| + LA loss ($\tau$=1) | 60.22 (-5.39) |
| + DASH | 65.62 (+0.01) |
| + CAP (Ours) | 89.50 (+23.89) |
| + CAI (Ours) | 88.15 (+22.54) |
| + CADR (Ours) | **93.79 (+28.18)** |

Table 3: Comparison with multiple baseline methods. Experiments are conducted on CIFAR-10 ($\gamma = N_{max} = 50$).

**More MNAR Settings.** In Figure 4, we show more different MNAR scenarios. Figure 4(a) shows the case where the unlabeled data is imbalanced. $\gamma_u$ denotes the imbalance ratio of unlabeled data,

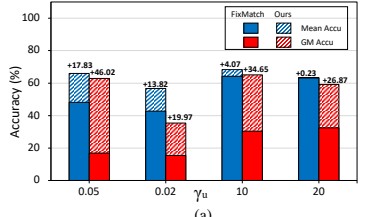

(a)

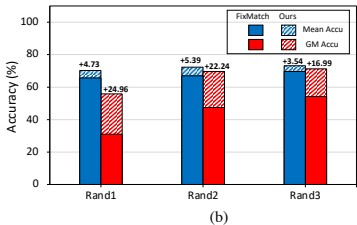

(b)

Figure 4: More MNAR settings. Evaluation of average accuracies (Mean Accu) (%) / geometric mean accuracies (GM Accu) (%) using our method and FixMatch when a) varying imbalance ratio $\gamma_u$ of the unlabeled data; b) selecting a sub-set randomly as labeled data. Experiments are conducted on (a) CIFAR-10 with $N_{max} = \gamma_l = 50$, and (b) CIFAR-100 with 5,000 labeled samples.

and those unlabeled data is inversely imbalanced (the rarest class in labeled data is the most frequent in unlabeled data) when $\gamma_u < 1$. Apart from the artificially designed imbalanced distribution, Figure 4(b) describes the results under the case where we label a random subset of data unaware of class distribution of the whole data. Without strictly selecting labeled data following the class distribution, the class distribution of labeled data can also be different from the unlabeled data. To show our effectiveness in producing unbiased predictions, we also report geometric mean accuracy (GM Accu) (Kubat, 2000; Kim et al., 2020). The overall results demonstrate that our method is robust to the distribution perturbation of both labeled and unlabeled data. Specifically, the significantly improved GM Accuracy shows that the class-wise accuracy obtained through our unbiased estimation is both high and in good balance.

**Compatibility with Conventional Settings.** As discussed in Section 3, our method degrades to the original FixMatch implementation when labels are missing completely at random (MCAR). Table 4 demonstrates the compatibility to MCAR of our methods: our results are consistent with the performance of FixMatch when labeled data are uniformly sampled from each class. Particularly, when provided with the extremely limited number of labeled data (CIFAR-10, #labels = 40), our performance is better than FixMatch (94.41% to 91.96%). The reason is that compared to FixMatch, our method is more robust to the data variance among mini-batches with the moving averaged propensity in such few-shot labeled data learning.

| | CIFAR-10 | | | CIFAR-100 | | | STL-10 |
|---|---|---|---|---|---|---|---|
| Method | #labels=40 | 250 | 4,000 | 400 | 2,500 | 10,000 | 1,000 |
| FixMatch | 88.61±3.35 | 94.93±0.33 | 95.69±0.15 | 50.05±3.01 | 71.36±0.24 | 76.82±0.11 | 94.83±0.63 |
| CADR (Ours) | 94.41 | 94.35 | 95.59 | 52.90 | 70.61 | 76.93 | 95.35 |

Table 4: Comparisons of average accuracies with labels missing completely at random (MCAR). Performances of Fixmatch are the reported results in their paper (Sohn et al., 2020). *#labels* denotes the overall number of the labeled data.

**Ablation Studies of Hyper-parameters.** We conducted ablation studies on two hyper-parameters: the moving average momentum $\mu$ for calculating class-aware propensity (Table 5), and the threshold scaling coefficient $\beta$ for class-aware imputation (Table 6). Table 5 shows the effectiveness of the moving average strategy comparing results of different momentum values, where "0" denotes no moving average. As shown in Table 6, when $\beta$ is too small (0.2), the requirement of confidence for rare classes lowers marginally, and thus the selected imputed labels still bear bias. Specifically, using the distribution of the labeled data to re-scale the threshold is not promising, while our method dynamically updating the threshold with training is adaptive and effective.

| $\mu$ | 0.999 | 0.99 | 0.9 | 0 |
|---|---|---|---|---|
| Accuracy(%) | 89.35 | 89.50 | 89.46 | 88.39 |

| $\beta$ | 0.2 | 0.5 | 1.0 | label |
|---|---|---|---|---|
| Accuracy(%) | 65.79 | 88.15 | 88.11 | 71.86 |

Table 5: Evaluation with varying moving averaging coefficient $\mu$ for CAP. Experiments are conducted on CIFAR-10 ($\gamma = N_{max} = 50$).

Table 6: Evaluation with different threshold coefficient $\beta$ for CAI. "label" means following the distribution of the labeled data. Experiments are conducted on CIFAR-10 ($\gamma = N_{max} = 50$).

## 6 CONCLUSION

In this work, we presented a principled approach to handle the non-random missing label problem in semi-supervised learning. First, we proposed Class-Aware Propensity (CAP) to train an improved classifier using the biased labeled data. Our CAP exploits the class distribution information of the unlabeled data to achieve unbiased missing label imputation. Second, we proposed Class-Aware Imputation (CAI) that dynamically adjusts the threshold in filtering pseudo-labels of different classes. Finally, we combined CAP and CAI into a doubly robust estimator (CADR) for the overall SSL model. Under various label missing not at random (MNAR) settings for several image classification benchmarks, we demonstrated that our method gains a significant and robust improvement over existing baselines. For future work, we are interested in incorporating 1) the selection bias theory (Bareinboim et al., 2014) of the missing data mechanisms and 2) the causal direction between label and data (Kügelgen et al., 2020), into semi-supervised learning.

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
