# OpenReview forum: "On Non-Random Missing Labels in Semi-Supervised Learning"
_ICLR.cc/2022/Conference — ICLR 2022 Poster_

### Official Review · Reviewer_9j41 · 2021-11-01

**Correctness:** 3
**Technical Novelty And Significance:** 3
**Empirical Novelty And Significance:** 3
**Recommendation:** 6
**Confidence:** 2

**Main Review:**

Attacking semi-supervised learning with the MNAR setting is an interesting and important topic for practical learning tasks. The empirical study is comprehensive and the proposed method provides a significant improvement compared to state-of-the-art semi-supervised learning approaches.

Weakness:
Two SOTA baselines of semi-supervised learning, MixMatch (https://proceedings.neurips.cc/paper/2019/file/1cd138d0499a68f4bb72bee04bbec2d7-Paper.pdf) and ReMixMatch (https://openreview.net/pdf?id=HklkeR4KPB) are not included in the empirical study.

**Summary Of The Paper:**

This work proposes a semi-supervised learning algorithm with systematic label missing, aka Missing Not At Random. The proposed solution is composed of two steps. First, to learn the classifier with the labelled data instances, a learnable weight, named propensity score, is proposed and attached to each labelled data instance to deliver an unbiased estimate (see Eq.10). Second, a class-wise threshold is tuned according to the class-specific propensity score (see Eq.11). The results show better semi-supervised classification accuracy compared to other baselines in the MNAR scenario.

**Summary Of The Review:**

Please find our comments above.

---

> ### Author Response · Authors · 2021-11-20
> **Response to Reviewer 9j41**
>
> We sincerely appreciate the reviewer for constructive suggestions to improve the paper. We thank the reviewer for acknowledging our methods and empirical studies as interesting and comprehensive. We tried our best to address the reviewer’s concerns, and hope the response can answer the following question.
>
> **Q: More Experiments on MixMatch and ReMixMatch**
>
> **A:** In the first version, we have included the comparisons of main results with MixMatch and ReMixMatch in Table R-4. Following the reviewer's suggestion, we report the ablation studies on MixMatch and RemixMatch in the following table:
>
>
> |            	| CIFAR 	| -10    	|        	| CIFAR  	| -100   	| STL-10 	| mini-ImageNet 	|
> |------------	|--------	|--------	|:--------:	|:--------:	|:--------:	|:--------:	|:---------------:	|
> |            	|   20   	|   50   	|   100  	|   100  	|   200  	|   100  	|      100      	|
> | MixMatch   	| 26.63  	| 31.28  	| 28.02  	| 41.32  	| 42.92  	| 28.31  	|     18.30     	|
> | w/ CAP     	| 40.34       	|  43.51      	|    45.47    	| 42.45  	| 46.54  	|  34.76 	|     22.09     	|
> | ReMixMatch 	| 41.84  	| 38.44  	| 38.20  	| 39.71  	| 39.22  	| 39.55  	|     23.50     	|
> | w/ CAP     	|  51.90      	|   55.03     	|  53.44      	|   40.15     	|   39.40     	|  42.53 	|     23.74     	|
>
> Table R-4: Comparison of mean accuracies (\%) with MixMatch and ReMixMatch.
> We alter the imbalance ratio $\gamma$ of labeled data and leave the unlabeled data balanced ($\gamma_u=1$). We keep $N_{max}=\gamma$ so that the least number of labeled data among all the classes is always 1.
>
> Since MixMatch and ReMixMatch used temperature-sharpened soft-labels rather than $argmax$ pseudo-labels in Fixmatch, they do not introduce a threshold in the label imputation process. Consequently, our CAI, which formulates an adaptive threshold, is not directly applicable, and we only applied CAP, which removes the bias through the proposed class-aware propensity. As shown in Table R-4, our method consistently boosts the performance, especially outperforming the baselines by large margins on CIFAR10 and STL-10. However, as MixMatch and ReMixmatch deliver severely inferior results than FixMatch in MNAR problems, i.e., more than 10\% gap of performance on CIFAR-10/100, we recommend applying our methods to FixMatch to achieve better results.

---

### Official Review · Reviewer_Fenu · 2021-11-02

**Correctness:** 4
**Technical Novelty And Significance:** 3
**Empirical Novelty And Significance:** 2
**Recommendation:** 6
**Confidence:** 3

**Main Review:**

Strengths

1. The paper does a good job of laying out the problem assumptions and differentiating from random labeling scenarios.

2. The proposed solution is clearly explained and justified.

3. The experimental results are well presented and the comparison across multiple different datasets/scenarios does a good job building the case for the proposed solution.


Weaknesses

1. Given the paper proposes the MNAR setting, it would help strengthen the justification/interest to point to a real dataset/setting that matches this assumption as the experimental results all are based on simulated class-dependent labeling. This is the major weakness of the paper to me as it weakens the justification/interest in the proposed problem.

2. The experimental results seem strong, however for all of these datasets the dependence of labeling on class is extremely high (or zero in the demonstration of consistency in the missing at random case). It would be interesting to see the performance comparisons with baselines for label dependence in the intermediate range. This is especially true given the synthetic nature of the missing labels making it hard to determine whether the proposed labeling dependence is reasonable/realistic.

**Summary Of The Paper:**

This paper formulates the problem of semi-supervised learning where the likelihood of an example being labeled is dependent on the class of the example. Given this assumption, the authors propose an approach to mitigate the resulting distribution shifts that occur in both the labeled and unlabeled data sets. Finally, experimental results are shown for the class-dependent labeling scenario, with the proposed algorithm outperforming standard SSL approaches that assume random labeling probability.


**Summary Of The Review:**

I am in favor of accepting this paper as the problem and solution are well written and the experimental results seem reasonable. The main weaknesses are the lack of an empirical example to justify the proposed setting and demonstration of the proposed approach on only simulated cases of MNAR data.

---

> ### Author Response · Authors · 2021-11-19
> **Response to Reviewer Fenu**
>
> We sincerely appreciate the reviewer for constructive suggestions to improve the paper. We are encouraged that the reviewer found our explanation and justification are clear, our experimental results are reasonable and well presented, and our paper is well written. We have conducted additional experiments as suggested, and we will include them in the revised manuscript. We hope our response can address the following concerns.
>
> **Q1: More Discussion on Real Data Setting.**
>
> **A1:** To apply our methods to a real-data setting, we conducted experiments on the subsets of iNaturalist [1], a real-world dataset comprised of the natural images and labels collected from a citizen science website.
> iNaturalist has two popular versions, iNaturalist-2018 for long-tailed recognition [2] and iNaturalist-2021 for nearly balanced data recognition [3]. As iNaturalist-2021 supplements iNaturalist-2018 with abundant additional data in their overlapped classes, these two versions can be naturally used for our MNAR setting in SSL. Specifically, we used iNaturalist-2018 as the long-tailed labeled set and iNaturalist-2021 as the more balanced unlabeled set. We selected $N$ classes from these original datasets to establish the subset, where the selected classes are distributed in various super-classes, *e.g.*, plants and insects. After removing the overlapped images, we sampled 50 images per class from iNaturalist-2021 for performance evaluation.
>
> The results of 20-class ($N=20$) and 50-class ($N=50$) subsets are reported in the following Table R-2.  It shows that our proposed methods outperform the baseline FixMatch by large margins, demonstrating the effectiveness of our methods in handling real-world imbalanced labeled data. Visualization of the data distribution and hyper-parameter settings will be shown in the revised Appendix.
>
> | Methods           	| 	|   $N$	|
> |------------	|:----:	|:-----:	|
> |            	| 20 	|   50  	|
> | Supervised 	| 25.3 	| 17.09 	|
> | FixMatch   	| 43.2 	| 47.24 	|
> | w/   CAP     	| 48.8 	| **51.32**|
> | w/   CAI     	| 47.5 	| 48.43 	|
> | w/o CADR   	| ***50.8***	| 49.48 	|
> | w/   CADR    	| **51.6**	| ***50.14*** 	|
>
> **Table R-2**: Comparisons of average accuracies (\%) on the iNaturalist-subsets between the fully-supervised method, FixMatch, and our methods. We marked the **best** and ***second-best*** accuracies.
>
>
>
>
> **Q2: More experiments on labeling dependence in the intermediate range.**
>
> **A2:**
> In the first version, we constructed experiments using relatively high imbalance ratios to simulate various MNAR scenarios. The imbalance ratios are widely used in long-tailed recognition tasks [4, 5]. Following the reviewer's suggestion, we conducted more experiments under various and smaller dependence of labeling on classes:
>
>
>
> |  **Methods**         	| 	|       	|       	|       **$\gamma$** 	|       	|       	|       	|        	|
> |----------	|:-------:	|:-------:	|:-------:	|:-------:	|:-------:	|:-------:	|:-------:	|:--------:	|
> | 	|   *1*   	|   *2*   	|   *5*   	|   *10*  	|   *20* 	|   *50* 	|  *100*  	|   *200*  	|
> | Fixmatch 	| 78.54 	| 76.77 	| 74.71 	| 70.53 	| 66.79 	| 59.13 	| 54.78 	|  50.62 	|
> | Ours (CADR)    	| 79.02 	| 77.71 	| 76.32 	| 74.37 	| 73.47 	| 70.06 	| 64.47 	| 63.30  	|
>
> **Table R-3**: Comparison of average accuracies (\%). We alter the imbalance ratio $\gamma$ of the labeled data in the intermediate range from 1 to 200, where $\gamma=1$ is the case where the label is missing completely at random. The experiments are conducted on CIFAR-100, and we keep $N_{max}=200$ across all settings.
>
> As shown in Table R-3, our method can consistently boost the performance of baseline FixMatch under different levels of label dependence. Besides, the improvement is more significant with larger $\gamma$, *i.e.*, the high dependence of labeling on class. This observation is reasonable since the baseline method gradually fails to handle the data distribution shift between the labeled and unlabeled data in challenging MNAR problems.
>
>
> **References:**
>
> [1] Van Horn, G., et al. ''The iNaturalist Species Classification and Detection Dataset.'' CVPR 2018.
>
> [2] iNaturalist Competition 2018, https://sites.google.com/view/fgvc5/competitions/inaturalist
>
> [3] iNaturalist Challenge 2021, https://sites.google.com/view/fgvc8/competitions/inatchallenge2021
>
> [4] Kaidi Cao, et al. ''Learning imbalanced datasets with label-distribution-aware margin loss.''  NeurIPS 2019.
>
> [5] Chen Wei, et al. ''CReST: A Class-Rebalancing Self-Training Framework for Imbalanced Semi-Supervised Learning.'' CVPR 2021.

---

### Official Review · Reviewer_D3Mi · 2021-11-04

**Correctness:** 4
**Technical Novelty And Significance:** 3
**Empirical Novelty And Significance:** 3
**Recommendation:** 8
**Confidence:** 4

**Main Review:**


### Strengths:

The paper made a clear review of the underlying theory for the pseudo-label-based semi-supervised learning framework. The framework can lead to an unbiased estimator if the unlabeled/labeled data status is fully random with respect to the overall data distribution.

CAP: The imbalanced availability of labeled data across categories is a practically important issue. The paper presents clear and convincing descriptions of this problem setup. It revised the framework in a theoretically decent way to incorporate the class-aware propensity. In other words, the paper proposed a probabilistically rigorous method to reweight each sample’s training loss adaptively. The implementation of this idea in SGD with mini-batches and running average is also reasonable.

CAI: The paper recognized the fixed threshold of FixMatch as a cause of bias across different categories. It proposed a formulation to set an adaptive threshold for each class based on the previous class-aware propensity. Though not very theoretically founded, the method agrees well with intuition (similar to the “focal loss” for object detection). The formulation can also gracefully fall back to the conventional FixMatch.

Using the theory of double robustness to combine CAI and CAP provides a theoretically better way to combine CAP and CAI than simply summing them together.

The experimental results in the synthetic class-imbalanced settings demonstrated the effectiveness of CAP and CAI. They outperformed FixMatch and other previous semi-supervised learning methods significantly. In the fully random settings for labeled data, the proposed methods also did not hurt the model’s accuracy; in fact, it improved the model’s performance a bit.

### Weakness:

In comparison to the trivial combination of CAP and CAI, CDAR did not always lead to stronger results (though CDAR showed significant improvement over the trivial combination in some cases). Is there any explanation for the slightly mixed performance of CDAR?

The paper can be stronger if it can show experimental results on a real-world dataset (i.e., the labeled data are not synthetically imbalanced but originally imbalanced).


**Summary Of The Paper:**

The paper addresses the semi-supervised learning problem under the setting of imbalanced label data by extending FixMatch with a class-aware method to reweight the loss and an adaptive thresholding method to select examples with pseudo labels. The two methods are termed Class-Aware Propensity (CAP) and Class-Aware Imputation (CAI), respectively. The paper also proposes the Class-Aware Doubly Robust (CADR) estimator by combining the two methods in the double robustness framework. Experiments are performed on Cifar-10/100, STL-10, and mini-ImageNet.

**Summary Of The Review:**

The paper proposes a novel method to tackle the more practical problem of semi-supervised learning with imbalanced data. The experimental results are strong, and the ablation studies are informative. More discussions can be made about CDAR’s performance.

---

> ### Author Response · Authors · 2021-11-19
> **Response to Reviewer D3Mi**
>
> We sincerely thank the reviewer for the insightful comments and suggestions. We are encouraged that the reviewer founds that our addressed problem is practical, our method is novel, and our results are informative. We tried our best to address the reviewer’s concerns, and the manuscript will be updated shortly to reflect any indicated changes. We hope the response can answer the following questions.
>
> **Q1: Explanation on that CADR does not always lead to stronger results compared to the trivial combination of CAP and CAI.**
>
> **A1**: We proposed Class-Aware Doubly Robust (CADR) estimator to improve the **robustness** of the performance. By robustness, we mean that the model can deliver **consistently** strong performance under different Missing Not At Random (MNAR) scenarios. Recall that the theoretical unbiasedness of the trivial combination requires both CAP and CAI to be unbiased. Differently, according to the theory of doubly robust methods, CADR maintains unbiasedness when either CAP or CAI is unbiased. Therefore, it is reasonable that CADR did not outperform the components or the trivial combination by large margins in all cases, especially when CAP and CAI are both unbiased, *i.e.*, the gap between the performances of CAP and CAI is small.
>
> According to the experimental results in Table 2, we found that our CADR is consistently among the best or second-best performance, agnostic to the value of $\gamma$ and the performance gap between individual components. When the label data is more imbalanced, *e.g.*, $\gamma=100$, CAP outperforms CAI by large margins on CIFAR-100 (63.07\% *vs.* 58.26\%), STL-10 (77.69\% *vs.* 71.21\%), and mini-ImageNet (32.23\% *vs.* 30.94\%), and our CADR outperforms the trivial combination by 0.52\% $\sim$ 1.26\%. The special case is $\gamma=100$ on CIFAR-10, where the trivial combination slightly beats CADR (93.97\% *vs.* 94.10\%), and the gap between CAP and CAI is also small (93.96\% *vs.* 93.86\%). These results empirically verified our theoretical analysis and the robustness of our CADR. To sum up, CADR can provide a reliable solution with robust performance and bring convenience to real applications.
>
>
> **Q2: Results on a real-world dataset.**
>
> **A2:** We thank the reviewer for the suggestion. To apply our methods to real-data MNAR occasions, we conducted experiments on the subsets of iNaturalist [1], a real-world dataset comprised of the natural images and labels collected from a citizen science website.
> iNaturalist has two popular versions, iNaturalist-2018 for long-tailed recognition [2] and iNaturalist-2021 for nearly balanced data recognition [3]. As iNaturalist-2021 supplements iNaturalist-2018 with abundant additional data in their overlapped classes, these two versions can be naturally used for our MNAR setting in SSL. Specifically, we used iNaturalist-2018 as the long-tailed labeled set and iNaturalist-2021 as the more balanced unlabeled set. We selected $N$ classes from these original datasets to establish the subset, where the selected classes are distributed in various super-classes, *e.g.*, plants and insects. After removing the overlapped images, we sampled 50 images per class from iNaturalist-2021 for performance evaluation.
>
> The results of 20-class ($N=20$) and 50-class ($N=50$) subsets are reported in the following Table R-1.  It shows that our proposed methods outperform the baseline FixMatch by large margins, demonstrating the effectiveness of our methods in handling real-world imbalanced labeled data. Visualization of the data distribution and hyper-parameter settings will be shown in the revised Appendix.
>
> | Methods           	| 	|   $N$	|
> |------------	|:----:	|:-----:	|
> |            	| 20 	|   50  	|
> | Supervised 	| 25.3 	| 17.09 	|
> | FixMatch   	| 43.2 	| 47.24 	|
> | w/   CAP     	| 48.8 	| **51.32**|
> | w/   CAI     	| 47.5 	| 48.43 	|
> | w/o CADR   	| ***50.8***	| 49.48 	|
> | w/   CADR    	| **51.6**	| ***50.14*** 	|
>
> **Table R-1**: Comparisons of average accuracies (\%) on the iNaturalist-subsets between the fully-supervised method, FixMatch, and our methods. We marked the **best** and ***second-best*** accuracies.
>
>
>
> **References:**
>
> [1] Van Horn, G., et al. ''The iNaturalist Species Classification and Detection Dataset.'' CVPR 2018.
>
> [2] iNaturalist Competition 2018, https://sites.google.com/view/fgvc5/competitions/inaturalist
>
> [3] iNaturalist Challenge 2021, https://sites.google.com/view/fgvc8/competitions/inatchallenge2021

---

### Author Response · Authors · 2021-11-20
**Summary of Paper Revision**

We gratefully thank all the reviewers for their valuable and constructive comments. We are encouraged that they find our topic interesting and important (Reviewer 9j41), our view for semi-supervised learning clear and theoretically decent (Reviewer D3Mi), our method probabilistically rigorous, theoretically effective (Reviewer D3Mi), and novel (Reviewer Fenu), and our explanation and justification clear (Reviewer Fenu). We are also glad that they acknowledge our implementation is reasonable (Reviewer D3Mi), our experimental results are strong and informative (Reviewer D3Mi, Reviewer Fenu), and our improvement is significant (Reviewer D3Mi).

We address the concerns and questions in detail below. According to these comments, we have improved our manuscript and summarized the main changes as follows:

1. Supplementary experiments on subsets of the real-world dataset iNaturalist (Appendix A.4.2, to Reviewer D3Mi and Fenu)
2. Revised discussion of the performance with our CADR (Section 5.2, to Reviewer D3Mi)
3. Supplementary comparisons under varying labeling dependence on the class (Appendix A.4.1, to Reviewer Fenu)
4. Supplementary ablation studies on more baselines (Appendix A.4.3, to Reviewer 9j41).

---

### Decision · Program_Chairs · 2022-01-20

**Decision:**

Accept (Poster)

**Comment:**

Addressed semi-supervised learning with the MNAR setting.  Well written paper.
Several additional experiments were reported in response to the reviewer questions.
General agreement amongst reviewers.